# Damage-programmable design of metamaterials achieving crack-resisting mechanisms seen in nature

Zhenyang Gao[1,2], Xiaolin Zhang[1,2], Yi Wu[1,2,3,4] ✉, Minh-Son Pham [5], Yang Lu [6], Cunjuan Xia[1,2,3,4], Haowei Wang[1,2,3] & Hongze Wang [1,2,3,4,7] ✉

The fracture behaviour of artificial metamaterials often leads to catastrophic failures with limited resistance to crack propagation. In contrast, natural materials such as bones and ceramics possess microstructures that give rise to spatially controllable crack path and toughened material resistance to crack advances. This study presents an approach that is inspired by nature's strengthening mechanisms to develop a systematic design method enabling damage-programmable metamaterials with engineerable microfibers in the cells that can spatially program the micro-scale crack behaviour. Machine learning is applied to provide an effective design engine that accelerate the generation of damage-programmable cells that offer advanced toughening functionality such as crack bowing, crack deflection, and shielding seen in natural materials; and are optimised for a given programming of crack path. This paper shows that such toughening features effectively enable crack-resisting mechanisms on the basis of the crack tip interactions, crack shielding, crack bridging and synergistic combinations of these mechanisms, increasing up to 1,235% absorbed fracture energy in comparison to conventional meta-materials. The proposed approach can have broad implications in the design of damage-tolerant materials, and lightweight engineering systems where significant fracture resistances or highly programmable damages for high performances are sought after.

With the advancement of fabrication technologies, metamaterials with complex microstructural geometries have been designed and realized, leading to the discovery of novel properties such as electromagnetic manipulation[1,2], thermal cloaking effects[3,4], and acoustic controls[5,6]. Mechanical metamaterials are a class of metamaterials with extraordinary mechanical properties, such as ultrahigh stiffness-to-weight ratio[7,8], ultrahigh energy absorption[9–12], negative Poisson's ratio[13,14], damage tolerance[15], and multistability[8,16–18]. By combining desirable

metamaterials with advances of manufacture[19], these metamaterials could provide promising solutions to replace conventional materials in high-value applications such as biomedical industries[20–22], aerospace[23–25], and civil engineering[26,27]. Recent studies have shown that enabling mechanical metamaterials to deform in programmable ways can significantly enhance their mechanical properties, where novel design strategies have been proposed to tailor the structures of metamaterials for pre-engineered compressive deformations[28,29] or

[1]State Key Labortory of Metal Matrix Composites, Shanghai Jiao Tong University, Shanghai 200240, China. [2]School of Materials Science and Engineering, Shanghai Jiao Tong University, Shanghai 200240, China. [3]Institute of Alumics Materials, Shanghai Jiao Tong University (Anhui), Huaibei 235000, China. [4]Anhui Province Industrial Generic Technology Research Center for Alumics Materials, Huaibei Normal University, Huaibei, Anhui 235000, China. [5]Department of Materials, Imperial College London, London SW7 2AZ, UK. [6]Department of Mechanical Engineering, University of Hong Kong, Hongkong 999077, China. [7]Shanghai Key Laboratory of Material Laser Processing and Modification, Shanghai 200240, China. ✉e-mail: eagle51@sjtu.edu.cn; hz.wang@sjtu.edu.cn

carefully engineer the crystal-mimicking phase meta-structures[30]. However, making such materials fracture in programmable manner to avoid catastrophic failure remains a challenge due to the stochastic nature of crack initiation, and fast propagation of fracture in high strength mechanical metamaterials[31-33]. The reduced relative densities[34,35], subtle features[36,37], and intricate geometrical features[38,39] make the mechanical metamaterials even more vulnerable to fracture. Existing research has mainly concentrated on generally deflect the crack by introducing pre-designed vacancies[40-42] or localized material weakening[43], while alternative methods attempted to enhance overall fracture energy through introducing nanolattices[44,45], multi-material designs[46], or new repetitive design structures[47,48]. Despite the increased complexity in the design of metamaterials, systematic design methods for precisely spatial damage programming are lacking, and effective crack-resisting mechanisms targeted for metamaterials have yet to be established. Consequently, the fundamental issue of unpredictable fractures and insufficient mechanisms to control and resist crack propagation has not been addressed.

Various materials such as human cortical bones[49-51], ceramics[52,53], layered rocks[54], and metal composites[55,56] possess complex hierarchical features ranging from micro to nano-scales to resist fracture (Fig. 1a). The main crack-resisting mechanisms observed in nature (Fig. 1b) include (1) crack tip interactions[57]: where the progress of a crack is hindered by the effective interactions between the crack tip and obstacles or deflectors along the fracture path; (2) crack shielding[58,59]: where the formation of a shielding zone that reduces the stress intensity at the crack tip, potentially blunting the crack tip and resisting further crack propagation; and (3) reinforcement bridging[60]: where tough fibrous ligaments maintain connection between fracture surfaces along the path of cracks to reduce the stress intensity and improve the toughness. Additionally, fracture guiding fibers or microcracks can direct the propagation of a crack along desired paths to sacrifice regions, further reducing the damage and protecting vulnerable and/or important regions. This prompts the question of whether it's possible to develop metamaterials with capability of spatially programming the damage and effectively resisting the crack propagation, akin to the damage-programming observed in materials from nature.

To translate the crack resistance and control mechanisms from nature to mechanical metamaterials, metamaterials with damage-programmable (DP) cells in micro-scale are created mimicking natural materials, possessing tailorable orientations of microfibers for programmable damage profiles (Fig. 1a). Enabled by machine learning (ML), a data-driven damage-programming metamaterial design method has been developed to achieve spatial control of damage propagation, and to program fracture toughening units (Supplementary Note 1–3, Supplementary Figs. 1–3). These units emulate the fracture properties of particles and phases forming the crack-resisting mechanisms observed in nature (Fig. 1b), serving as the building blocks to construct next-generation metamaterials containing advanced crack-resisting mechanisms. The present study examines the role of such units in the toughening of metamaterials, providing a foundation to the development of lightweight materials with the capability to precisely control and arrest the damage (see Supplementary Note 3 and Supplementary Fig. 3 for the rational of the building blocks). It is demonstrated that DP metamaterials exhibit significantly enhanced fracture resistance, absorbing up to 1235% more fracture energy per unit propagation compared to conventional metamaterials. The functional programming of the damage also allows the development of engineering systems with fracture-guiding functions to enhance the safety and protection.

## Results and discussion
### Spatially programming the damage of mechanical metamaterials
Materials found in nature often exhibit controlled fracture behavior due to the presence of microscopic structures such as microcracks and microfibers. Inspired by this, the data-driven damage-programming metamaterial design method is initially used to develop DP metamaterials with spatially programmable three-dimensional (3D) damages through crack path engineering (see "Methods" section and Supplementary Note 2 for the details of materials and crack path engineering designs). The data-driven damage-programming metamaterial design method includes a ML model (Supplementary Fig. 2b) that learned significant datasets of fracture strength ($\sigma_f$), fracture energy ($G_f$), and fracture angle ($\theta_f$) for DP design, and used a ML-enabled DP cell generation algorithm to find optimal fiber orientations of DP cells to achieve desired ($\sigma_f$, $G_f$, $\theta_f$) values (refer to Supplementary Note 1, Supplementary Fig. 4, and Supplementary Fig. 5 for the algorithmic details, model validations, computational cost of the training, data prediction, and ML-assisted DP cell generation). For illustration purposes, the topology of the base architecture was a body-centered cubic (BCC) structure; however, the proposed DP method is not limited to this base topology.

The effectiveness of the proposed damage-programming designs was experimentally demonstrated through programming of a trigonometric fracture surface (Fig. 2a, see Supplementary Fig. 6a, b and "Methods" section, Supplementary Note 4, and Supplementary Fig. 6 for test configurations, morphology designs of crack geometry, and a further experimental validation, respectively). The experimental examination indicates that the targeted crack path was achieved as design (Fig. 2b, and Supplementary Movie 1, also see "Methods" section), confirming the validity and effectiveness of the proposed method. DIC data in Fig. 2c indicates a significant fracture confinement effect initiated at the programmed crack path prior to the crack entering, where the strain confinement increases up to 94% until the complete fracture of the guiding cell. This was consistent with the designed behavior of guiding cells, which are more deformable before crack entry, and guide the crack to targeted regions. It is also noteworthy that the proposed method can be straightforwardly applied to more desirable and complicated 2D and 3D shapes based on given fracture design targets (Supplementary Note 4).

### Enabling nature's toughening mechanisms in metamaterials
In nature, tough materials often contain micro-scale toughening units that enable crack-resisting mechanisms including crack tip interactions, crack shielding, and reinforcement bridging[61-63] (Fig. 2a) and Supplementary Fig. 3). The toughening units enabling different crack-resisting mechanisms are summarized in Table 1. Specifically, crack tip interactions are typically triggered by crack front bowing (CB) and crack deflection (CD) phases, which hinder crack propagation and deflect the crack directions. Crack shielding is achieved through shield units that blunt crack initiation, while reinforcement bridging incorporates ultra-stiff reinforcement bridges that impede crack extension. Here, artificial toughening units are constructed by the data-driven damage-programming design method including CB and CD phases, shield units, and reinforcement bridges using combinations of functionally programmed DP cells (Fig. 3a (middle), also see Supplementary Note 3 for the detailed designs and rationales of DP cells for toughening units). The micro-scale 3D crack propagations observed by X-ray computed tomography (XCT) experiments of the toughening units closely replicate the programmed fracture schematics in Fig. 3a (left) (see "Methods" section for details of XCT analysis). This observation validates the successful activations of micro-scale fracture toughening events inspired from nature. The fracture energies normalized by densities of specimens programmed with those toughening units are experimentally studied (Fig. 3b), where the energy of each individual fracture toughening event is also numerically analyzed to understand the underlying sciences (see Supplementary Note 5 and "Methods" section for experimental details and simulation configuration, also see Supplementary Fig. 8 to Supplementary Fig. 16 for numerical results). The results in Fig. 3b demonstrate that

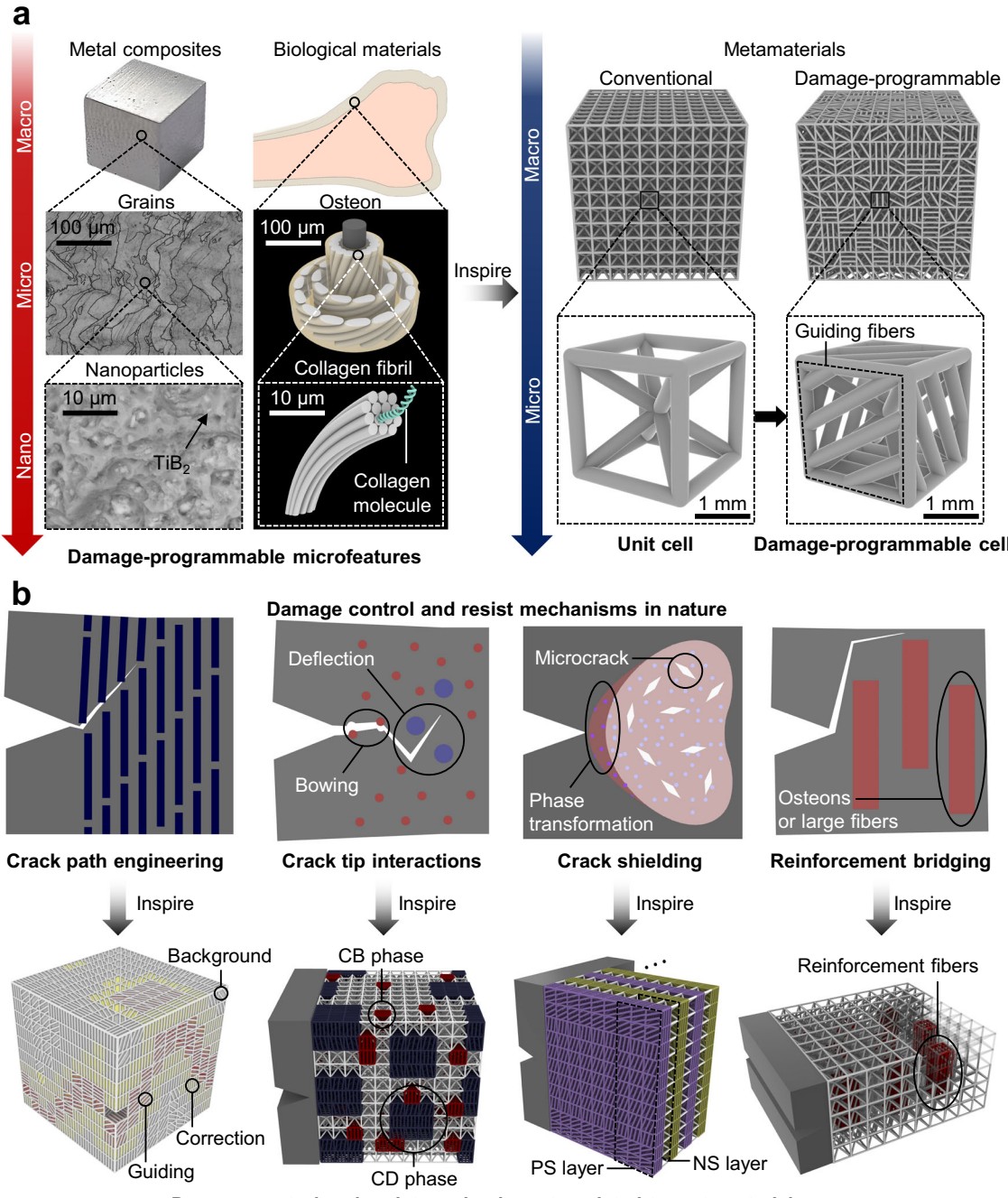

**Fig. 1 | The damage-programmable (DP) design of metamaterial inspired by nature. a** Toughening units that control the fracture behavior across length scales from micro to nano-scale in metal composites, biological materials, and metamaterials. A cubic sample of Al-Cu-Mg metal containing 2 wt.% of TiB2 particleswith their grains and nanoparticles were observed by scanning electron microscope (SEM) TiB$_2$. Schematic diagram of biological materials are modeled and rendered by Rhino 6. **b** DP metamaterial design strategies that enable the spatial programing of damage (the guiding, correction, and background zones are colored by red, yellow, and white, respectively) and crack-resisting mechanisms such as the crack tip interactions, crack shielding, and reinforcement bridging (reinforcement fibers colored by red), where the CB represents the crack bowing phase (red), the CD is the crack deflection phase (blue), PS (purple) and NS (yellow) are the positive and negative shielding, respectively. The terms "positive" and "negative" refer to the direction of crack propagation being guided either upwards or downwards, respectively, in relation to the initial crack direction. In the schematics, the blue squares in crack path engineering represent the hard fibers that alter the crack directions. The blue and red circles in crack tip interactions denote the large crack deflection phases and the small obstacles triggering crack bowing, respectively. In crack shielding, the dark blue circles, light blue circles, and white diamonds signify the particles without phase transformation (in the dark red zone), particles with phase transformation (in the light red zone), and microcracks, respectively. The red squares in reinforcement bridging represent the osteons or large fibers.

implementation of such toughening mechanisms considerably enhance the normalized fracture energies of metamaterials in comparison with ones without toughening units.

By constructing the toughening units to activate key toughening mechanisms, this paper studies the fracture mechanics of DP metamaterials with crack tip interactions, crack shielding, and reinforcement bridging (Fig. 3a (right)). Design parameters for the building blocks are selected based on fracture energy modeling (Supplementary Note 6). The effectiveness of the method is demonstrated by comparing the fracture energies of specimens with and without crack-

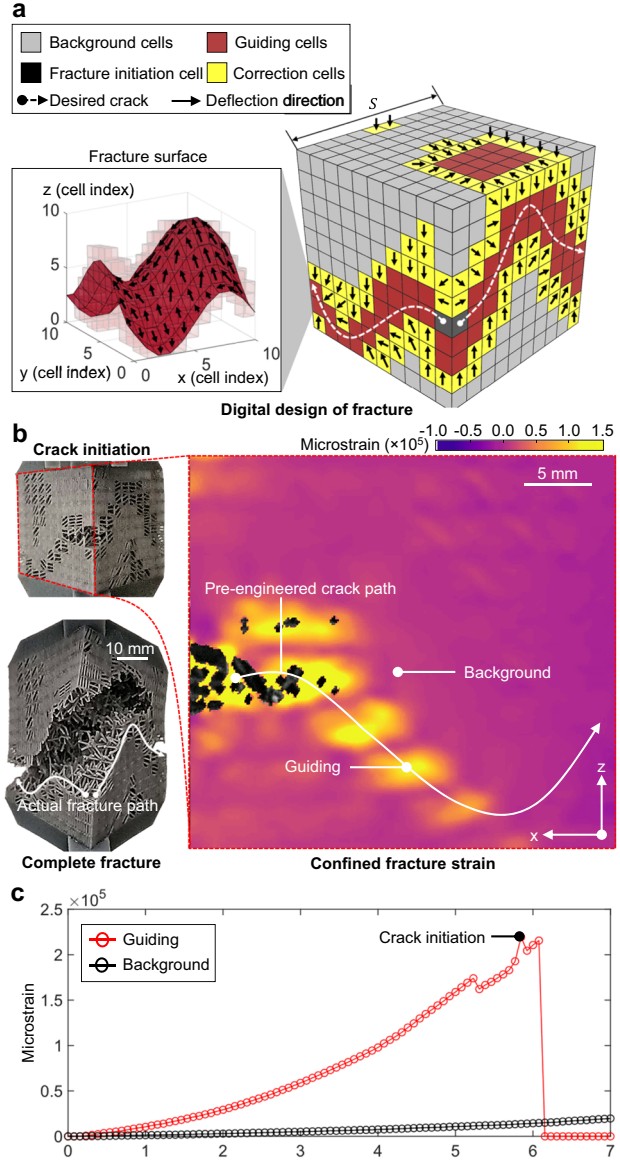

**Fig. 2 | Fracture path engineering using the data-driven damage-program-mable (DP) metamaterials. a** The design of DP metamaterials based on the desired fracture geometries, where $f_{3D}(x,y)$ represents the function of the 3D crack geometry defined in Supplementary Note 4, $S$ is the size of the 3D design space and test metamaterial chunk. **b** Experimentally observed fracture path of the DP metamaterials and the digital image correlation (DIC) analysis of the controlled fracture propagation. The white arrows indicate the visualized overall crack path according to the location of fractured DP cells on the front surfaces of the sample, based on experimental captures (on the sample image) and strain field (on the DIC result). The DP metamaterial images are captured by camera during the fracture experiments. **c** The strains in the guiding and background regions of the DP metamaterials. Source data are provided as a Source Data file.

resisting mechanisms, as indicated by the normalized fracture load-displacement data (Supplementary Fig. 17). It shows that the introduction of crack tip interactions, crack shielding, and reinforcement bridging to metamaterials increased, respectively, by 127%, 207%, and 122% of the normalized fracture energy, demonstrating the potential of translating the crack-resisting mechanisms from nature to mechanical metamaterials.

To gain further insight into the fracture behavior of the specimens, their fracture characteristics are analyzed at different stages of crack (Fig. 3a (right) and c). Per unit propagation along the direction of

**Table 1 | Summary of toughening units for crack-resisting mechanisms**

| Crack-resisting mechanism | Toughening unit(s) |
|---|---|
| Crack tip interaction | Crack bowing (CB) phase |
| | Crack deflection (CD) phase |
| Crack shielding | Shield unit |
| Reinforcement bridging | Reinforcement bridges |

the crack opening, the crack shielding specimen showed significant improvement in normalized fracture energy and strength (4.2 and 2.0 times compared to a BCC lattice metamaterial) in crack initiation due to the blunting effect thanks to the crack shielding (Supplementary Note 7 and 8), leading to a noticeable plateau in the peak load. However, a fast crack propagation occurs once the crack leaves the shielded region, resulting in a sudden drop in the normalized load-displacement curve. Designs with crack tip interactions showed moderate improvement in the normalized fracture energy of crack initiation thanks to crack deflection, and a significantly enhanced normalized fracture energy with a delayed fracture strain (3.2 and 1.9 times of the BCC metamaterial) during the crack propagation thanks to the CD and trapping. Similarly, the specimen with reinforcement bridges exhibited a slightly increased crack initiation strength and significant improvement in normalized fracture energy (1.2 and 2.2 times of the BCC design) thanks to three bridged deflection events. Our results suggest that while crack shielding is most beneficial in the crack initiation stage, crack tip interactions and reinforcement bridging demonstrate significant benefits in the crack propagation stage.

Theoretical models were proposed to interpret fracture energies contributed by different crack-resisting mechanisms, inspired from the fracture mechanics for tough materials in nature. It is discovered that the fracture energies ($G_f$) of the proposed DP metamaterials can be decomposed into the energy contributed by its matrix ($G_{f,BCC}$), and different types of energy barriers to crack advance, such as crack by-pass, crack cut-through, deflected crack, shielded crack, shielded deflection, and bridged crack (see Table 2 for the correspondence relationship between each term and its fracture toughening event, also see Supplementary Note 6 to 11 for experimental and numerical observations, and their detailed theoretical derivations):

$$G_{f,crack\,tip\,interaction} = G_{f,BCC} + \alpha_{CB}\rho_{CB}[\Delta_{CB1}G_{f0} + (\Delta_{CB1} + \Delta_{CB2})P_{CB2}G_{f0}] + \alpha_{CD}\Delta_{CD}l_f G_{f0} \tag{1}$$

$$G_{f,crack\,shielding} = G_{f,BCC} + \alpha_S(\Delta_S G_{f0} + P_{SD}\Delta_{SD}G_{f0}) \tag{2}$$

$$G_{f,reinforcement\,bridging} = G_{f,BCC} + \alpha_{B1}\Delta_{CD}O_{BD}G_{f0} + \alpha_{B2}\Delta_{CB2}O_{BC}G_{f0} + \alpha_{B3}\Delta_B A_B S_B G_{f0} \tag{3}$$

where $\alpha_{CB}$, $\alpha_{CD}$, $\alpha_s$, $\alpha_{B1}$, $\alpha_{B2}$, and $\alpha_{B3}$ are the geometrical scaling factors related to the size of the specimens; $\rho_{CB}$ is the density of the CB phase; $P_{CB2}$ is the scaled possibility of the cut-through event derived by the statistical modeling of agglomeration events; $l_f$ is the actual length of the fracture path; $P_{SD}$ is the potential function of the shielded deflection derived by the angle compliance analysis; $O_{BD}$ and $O_{BC}$ represents the occurrences of the bridged deflection and bridged cut-through, respectively; $A_B$ is statistical bridging area per length of the bridging fibers; $S_B$ represents the size of the reinforcement bridges; $G_{f0}$ is the energy barrier required for the crack to propagate through a base cell without FG features; $\Delta_{CB1}$, $\Delta_{CB1} + \Delta_{CB2}$, $\Delta_{CD}$, $\Delta_S$, $\Delta_{SD}$, $\Delta_B$ are the energy increment for the crack-resisting events by-pass, cut-through,

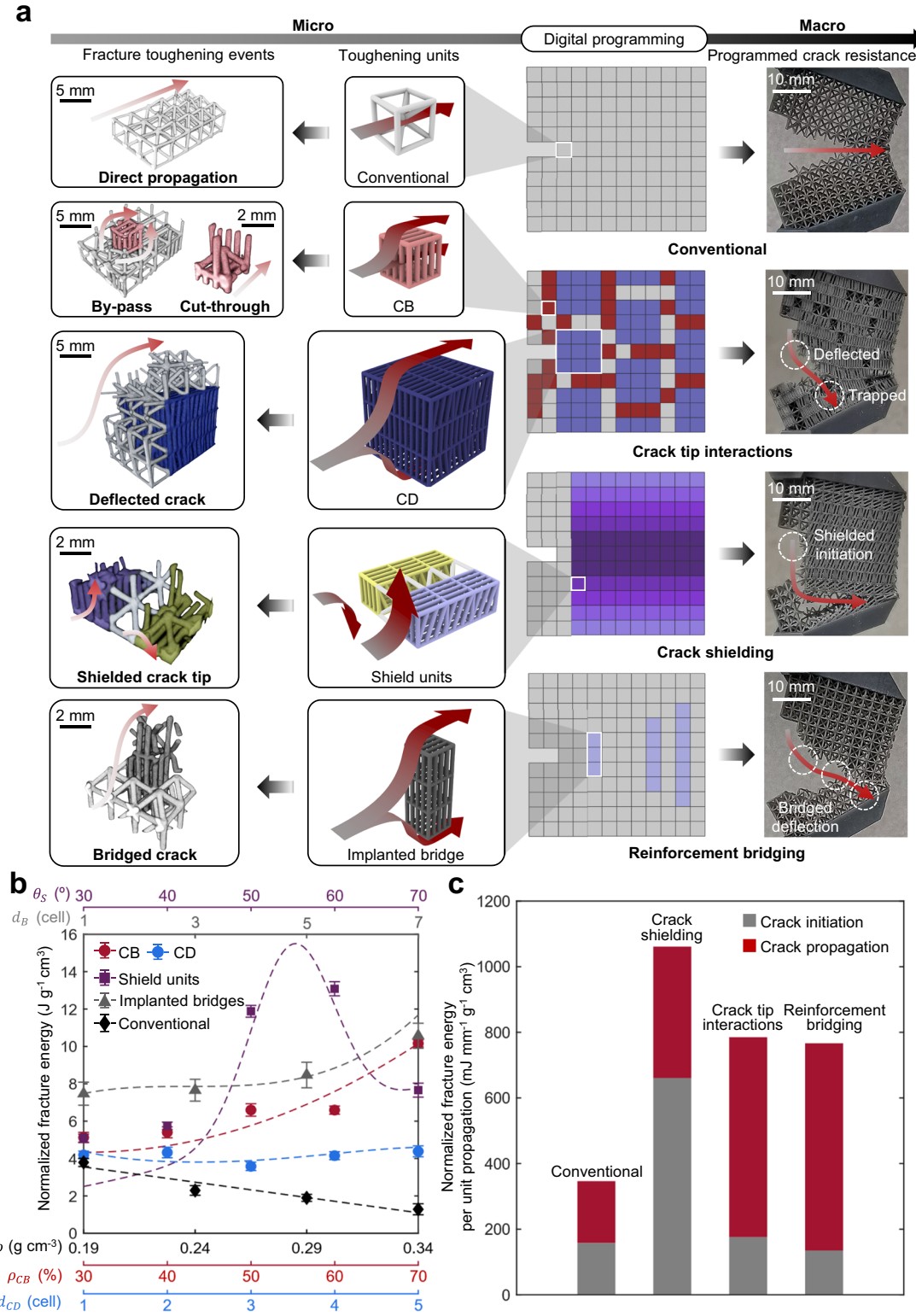

deflected crack, shielded crack tip, shielded crack with deflection, and bridged crack, respectively.

This theory is successful in describing the experimental results (Fig. 3b) that show the crack paths were well controlled as programmed, paving the way for designing DP metamaterials with crack tip interactions, crack shielding, and reinforcement bridging. The reported data also demonstrates a successful translation of natural toughening mechanisms into mechanical metamaterials, thereby establishing a foundation to program cracks in metamaterials.

## Damage-programmable metamaterials

In-depth examinations of crack resistance in metamaterials from the previous sections pave the way to develop metamaterials containing combined toughening features including crack tip interactions, crack shielding, reinforcement bridging, and crack dissipation resisting cracks at all stages of crack development (Fig. 4a). In particular, we applied crack shielding at the crack initiation region to enhance the crack initiation resistant strength. CD phases and reinforcement bridges were embedded to deflect the crack to regions filled with 50%

**Fig. 3 | Programming the fracture resisting features in mechanical metamaterial. a** Artificial toughening units inspired by nature (middle), micro-scale fracture toughening events observed using X-ray computed tomography (XCT) scanning (left), and their experimentally captured macroscale crack-resisting behaviors (right). Crack bowing (CB, red) phase is designed with type-1 DP cells, while crack deflection (CD, blue) phase is developed with a combination of type-1 and type-2 DP cells. Negative shielding (NS, yellow on the middle/left) and positive shielding (PS, purple on the middle/left) layers of the shield units (gradient purple on the right) are developed by type-3 DP cells. Light blue boxes on the right and dark gray structures on middle/left indicate reinforcement bridges, and light gray represent conventional cells without damage programming. Refer to Supplementary Note 3 for detailed design rationales of the toughening units and crack-resisting mechanisms. The images of metamaterial samples (right) are captured by camera during the fracture experiments. The red gradient arrows indicate the crack propagations according to XCT-scanned micro-scale fracture toughening events (left), schematic crack propagations of micro-scale fracture toughening events (middle), and the overall macroscopic fracture paths of the metamaterial samples (right). **b** The theoretical curves and experimentally validated fracture energies normalized by densities of conventional metamaterials and DP metamaterials with CB phases, CD phases, shield units, and reinforcement bridges, respectively. $\theta_s$, $d_B$, $\rho$, $\rho_{CB}$, and $d_{CD}$ represent shield angle, distance between adjacent reinforcement fibers, density of conventional metamaterials, density of CB phases, distance between the adjacent CD phases, respectively. The error bars are calculated based on the standard deviations from the results of three repetitive experiments. Note that certain error bars were small, making them almost indiscernible. **c** Quantification of normalized fracture energy per propagation unit associated with individual toughening mechanisms. Source data are provided as a Source Data file.

CB phases to trap the crack. A metamaterial without crack-resisting features with same beam diameters was also design as a reference for comparison. A void cell is included at the end of both the metamaterials to evaluate the protection of a potential vulnerable but important site. The metamaterial with crack-resisting features effectively diverts the crack away from this vulnerable site. The experimental results demonstrate that the DP metamaterial exhibits a significant improvement in toughness with a twofold increase in normalized fracture strength and significantly higher plateau load of fracture than those of the reference metamaterial. The reported data confirms the crack-resisting features in DP metamaterials effectively resist, arrest, and dissipate the cracks.

DIC and XCT experiments were used to study the role of crack-resisting features in spatial deformation and 3D crack propagation (Fig. 4b, c and Supplementary Movie 2, also see "Methods" section and Supplementary Fig. 18 for details of DIC experiments and XCT analysis). The DIC results reveal that the DP metamaterial exhibits a much more uniformly distributed strain field, with only one third of the maximum strain observed at the crack tip compared to the reference metamaterial (Supplementary Fig. 19a), resulting in crack tip blunting and increased fracture strength, as depicted in Fig. 4a. During crack propagation, the cracks are guided to two trapping regions with crack tip interactions. The strain fields of the two trapping regions compete, leading to the arrest of the initial crack and delayed initiation of a secondary crack, causing remarkable increase in crack propagation energy. In contrast, the reference metamaterial exhibited a highly localized strain field, leading to a rapid crack growth thereby a premature failure. For the protection of the vulnerable site represented by the void cell, the DIC results (Supplementary Fig. 19b) showed that the DP metamaterials provided excellent protection to the site which was deformed 84% less than in the reference metamaterial. These observations were further verified by 3D XCT characterizations that shows the shielded crack initiation, deflected and trapped crack propagation, and dissipated crack were achieved by the combined crack-resisting mechanisms, greatly improving the fracture energy absorption in comparison to fast and sharp cracks seen in a conventional metamaterial (Fig. 4c).

Figure 5a present the fracture energy per unit propagation of DP metamaterials developed in this study, as well as those of conventional metamaterials fabricated using the same parent materials, printing technology, and testing method (detailed in "Methods" section). The results reveal that FCC, BCC, octet-truss, vintiles, and tesseract metamaterials display limited fracture energy density (per crack propagation length unit along the crack opening direction), while the introduction of crack-resisting mechanisms, such as crack tip interactions, in DP metamaterials leads to an more than five-fold increased fracture energy density. The DP metamaterials display a remarkable fracture energy density up to 1,235% higher than that of conventional lattices at the same density (Fig. 5b). These results validate the effectiveness of translating the crack-resisting mechanisms inspired from nature to significantly enhance the fracture energy absorption capacity and provide ways to precisely control the spatial damage in mechanical metamaterials. This advancement opens opportunities for the development of next-generation lightweight metamaterials tailored for load-bearing applications with advanced crack resistance functions (Fig. 6a), and designable crack patterns (Fig. 6b and Supplementary Fig. 20).

In conclusion, the study presents a way to develop damage-programmable mechanical metamaterials by introducing fracture-controlling and crack-resisting features observed in natural materials to the artificial domain (Fig. 1). Detailed experimental examination demonstrates an achievement of precisely programming complex crack paths that had not been reported before (Fig. 2). Multiple crack-resisting features were studied and programed to enhance the crack tip interactions, crack shielding, and reinforcement bridging to create damage-programmable metamaterials. By strategically programming a variety of nature-inspired crack-resisting and dissipation mechanisms that are activated at different stages of crack propagation, substantial increases in resistance to crack advances and in fracture energy are achieved, up to 1335%, while also effectively dissipating the crack at desired locations, in comparison to metamaterials without such mechanisms (Figs. 4 and 5). The design of such damage-programmable metamaterials offers a promising avenue for the development of next-generation, lightweight engineering systems with fracture-controlling functions that might find uses in a range of applications such as the lightweight damage-programming bodies of

**Table 2 | Summary of energy barriers for different fracture toughening events**

| Observed fracture toughening events | Energy barrier |
|---|---|
| Crack by-pass | $\alpha_{CB}\rho_{CB}\Delta_{CB1}G_{f0}$ |
| Crack cut-through | $\alpha_{CB}\rho_{CB}\Delta_{CB2}P_{CB2}G_{f0}$ |
| Crack deflection | $\alpha_{CD}\Delta_{CD}l_fG_{f0}$ |
| Shielded crack | $\alpha_S\Delta_SG_{f0}$ |
| Shielded deflection | $\alpha_SP_{SD}\Delta_{SD}G_{f0}$ |
| Bridged deflection | $\alpha_{B1}\Delta_{CD}O_{BD}G_{f0}$ |
| Bridged cut-through | $\alpha_{B2}\Delta_{CB2}O_{BC}G_{f0}$ |
| Bridged crack | $\alpha_{B3}\Delta_BA_BS_BG_{f0}$ |

$\alpha_{CB}$, $\alpha_{CD}$, $\alpha_S$, $\alpha_{B1}$, $\alpha_{B2}$, and $\alpha_{B3}$ are the geometrical scaling factor related to the sizes of test specimens for the additional fracture energies caused by crack bowing, crack deflection, shielded crack, bridged deflection, bridged cut-through, and bridged crack, respectively. $\rho_{CB}$ is the density of crack bowing (CB) phase. $\Delta_{CB1}$, $\Delta_{CB2}$, $\Delta_{CD}$, $\Delta_S$, $\Delta_{SD}$, $\Delta_B$ are the fraction of energy increment for crack by-pass, crack cut-through, crack deflection, shielded crack, shielded deflection, and bridged crack, respectively. $G_{f0}$ represent the required fracture energy for propagating through a base cell. $P_{CB2}$ and $P_{CB2}$ are the scaled possibility of the cut-through event and the potential function of the shielded deflection, respectively. $l_f$ is the actual length of the deflected fracture path. $O_{BD}$ and $O_{BC}$ are the occurrences of the bridged deflection and bridged cut-through, respectively. $A_B$ is the statistical result of the bridging area per length of the bridging fibers. $S_B$ represents the size of the bridging fiber.

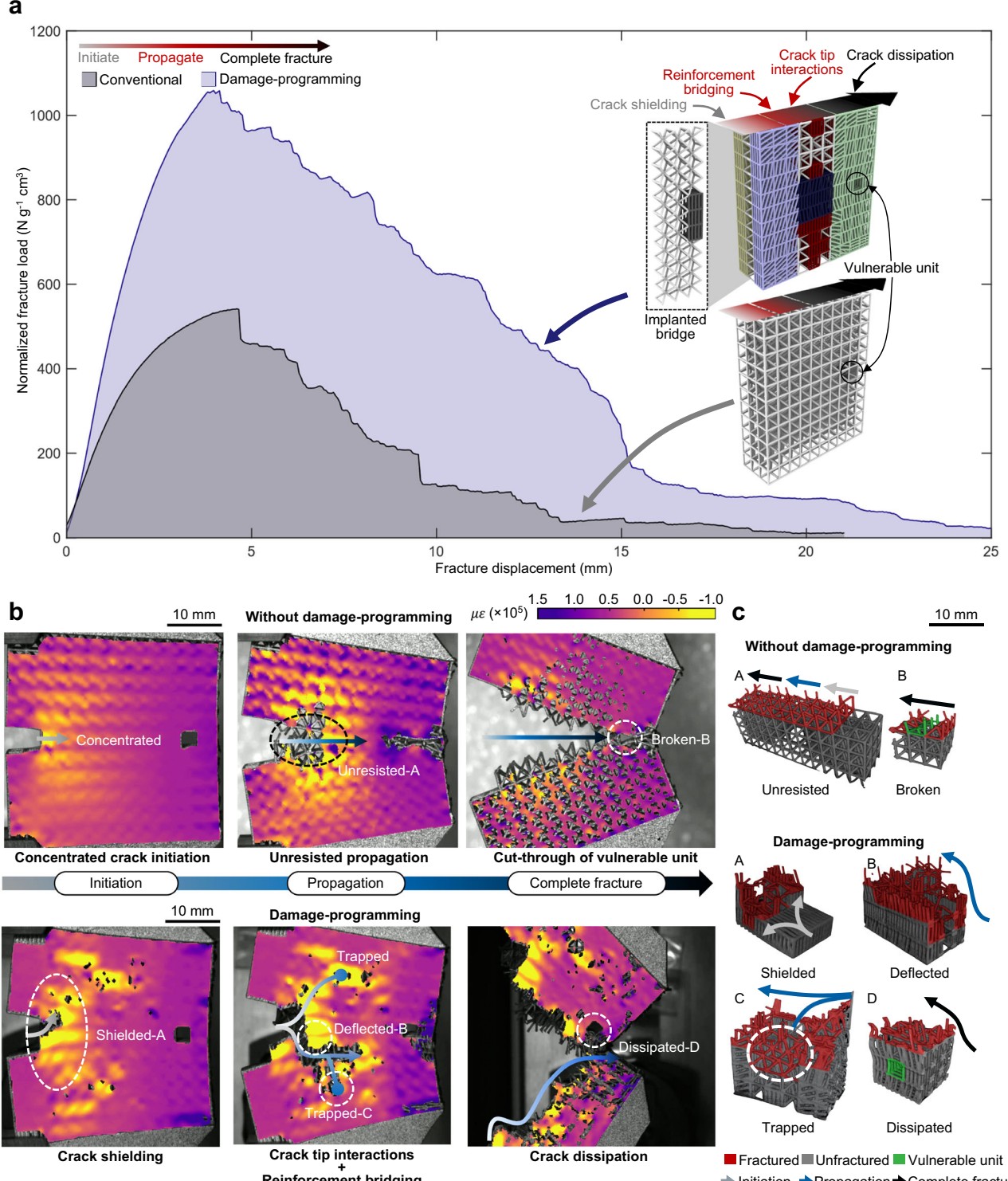

**Fig. 4 | Fracture of damage-programmable (DP) metamaterials containing all key toughening mechanisms. a** The fracture load-displacement curves normalized by densities and design schematics showing the DP metamaterials with all the combined crack resistance mechanisms in comparison to a conventional metamaterial with same beam diameters (used as a reference), where yellow, purple, blue, red, dark gray, and green represent the negative shielding layer, positive shielding layer, crack deflection phase, crack bowing phase, implanted bridge, and crack dissipation zone, respectively. **b** The digital image correlation (DIC) analysis of the crack initiation, propagation, and complete fracture for the DP metamaterials with the combined crack resistances versus the reference. **c** X-ray computed tomography (XCT) showing the micro-scale 3D crack propagation of metamaterials with and without damage-programming. Source data are provided as a Source Data file.

flying vehicles to enhance the protection of passengers against damages (Supplementary Fig. 20). The potential of our work extends far beyond the examples presented here. The approach proposed in the present study necessitate the needs of new research to obtain new insights into fracture behavior of such programmable metamaterials under complex fracture load scenarios (including complex load paths, high strain rates) and with different parent materials or meta-structure topologies.

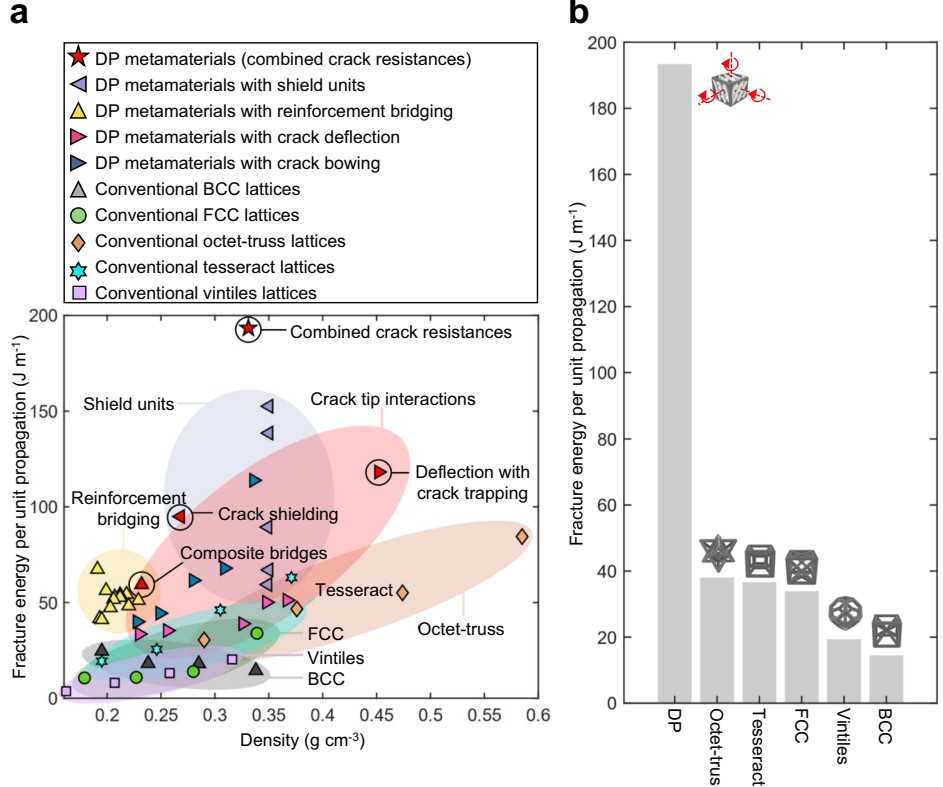

**Fig. 5 | Fracture properties of metamaterials measured using damage-programmable (DP) designs and existing designs including body-centered-cubic (BCC)[66], face-centered-cubic (FCC)[67], octet-truss[43], tesseract[68], and vintiles[69].** a The material property chart reflecting the density and the fracture energy per unit propagation for the DP metamaterials and conventional lattice structures. **b** The fracture energy per unit propagation for DP metamaterials with combined crack resistance mechanisms in comparison with conventional metamaterials of various lattice topologies with the same density. Source data are provided as a Source Data file.

## Methods

### Materials and experiments

All samples are prepared using Formlabs® Tough 2000 resin with Formlabs® Form 3 stereolithography (SLA) 3D printer with a resolution of 25 μm, 25 μm, and 50 μm in x, y, and z directions, respectively. Specimens are ultrasonic-washed with ethanol absolute solution, cured in an 80 °C ultraviolet environment for 120 min, and secured in a dark condition before the mechanical tests. The mechanical properties of the parent materials are evaluated according to ASTM D638 standard using ZwickRoell Z100 universal tensile test machine. The experimental results (Supplementary Table 2) suggest that the Formlabs® Tough 2000 material with the post-processing conditions in this study exhibits a modulus of 1.3 GPa, an ultimate tensile stress of 36.3 MPa, and an ultimate tensile strain of 35.4%.

To evaluate the fracture behaviors of metamaterials, three types of specimens are designed (Supplementary Fig. 6) and summarized in Supplementary Table 4. The three-layer configuration is applied to enable the evaluations of crack-resisting mechanisms with specifically extended design space vertical to the surface of the fracture propagations, such as the specimens that possess large 3 × 3 × 3 CD phases for the crack deflections. Depending on the maximum load, the thicknesses of the sidebars are designed according to the feasible design areas determined by Eqs. (4–5) (Supplementary Fig. 6c) to restrict the maximum bending displacement of the sidebars below 1 mm, which ensures that all specimens possess limited unexpected strains from the sidebars without wasting additional printing materials:

$$F(x) = F_{max} \exp(-0.15x) \tag{4}$$

$$\delta(x) = F(x)x^2 / (3EI(x)) \tag{5}$$

where $F(x)$ is the approximate load applied on the sidebars at position $x$ based on the simplified bending model, $F_{max}$ is the estimated peak load for a given specimen, $E$ is the modulus of the parent material, $I(x)$ is the moment of inertia of the side bar geometry at position $x$, $\delta(x)$ is the estimated bending displacement of the sidebar at position $x$. A pair of metal rods are inserted into the holding spaces of the sidebars and the fixtures to provide vertical fracture displacements from load cells with a load rate of 1 mm minute$^{-1}$.

### Finite element analysis

This paper applied FEA approach to obtain the fracture properties of the training data, and the local fracture behaviors of different crack-resisting mechanisms. The models of DP metamaterials are generated by the proposed design algorithms through the rhinopythonscript of Rhino 6[64], and imported as beam elements to Abaqus CAE[65] for FEA. Configurations of different types of simulations are provided in Supplementary Fig. 21, where the beams are decomposed to B31 mesh segments with sizes of 0.1. For the simulations of training data gathering, a pair of rigid rods are tied with the frontal beams of the DP cell to apply a constant displacement rate. For the microstructural fracture simulation of different crack-resisting mechanisms, the rigid plates are tied on the top and bottom layer of beams to simulate the local fracture process since the strain of the surrounding cells far from the crack tip is neglectable. The fracture displacement is applied on the front edges of the rigid plates, while the cell at the crack initiation position is deleted. General contacts are used as the interaction conditions during the simulations throughout this paper to simulate potential contacts of the

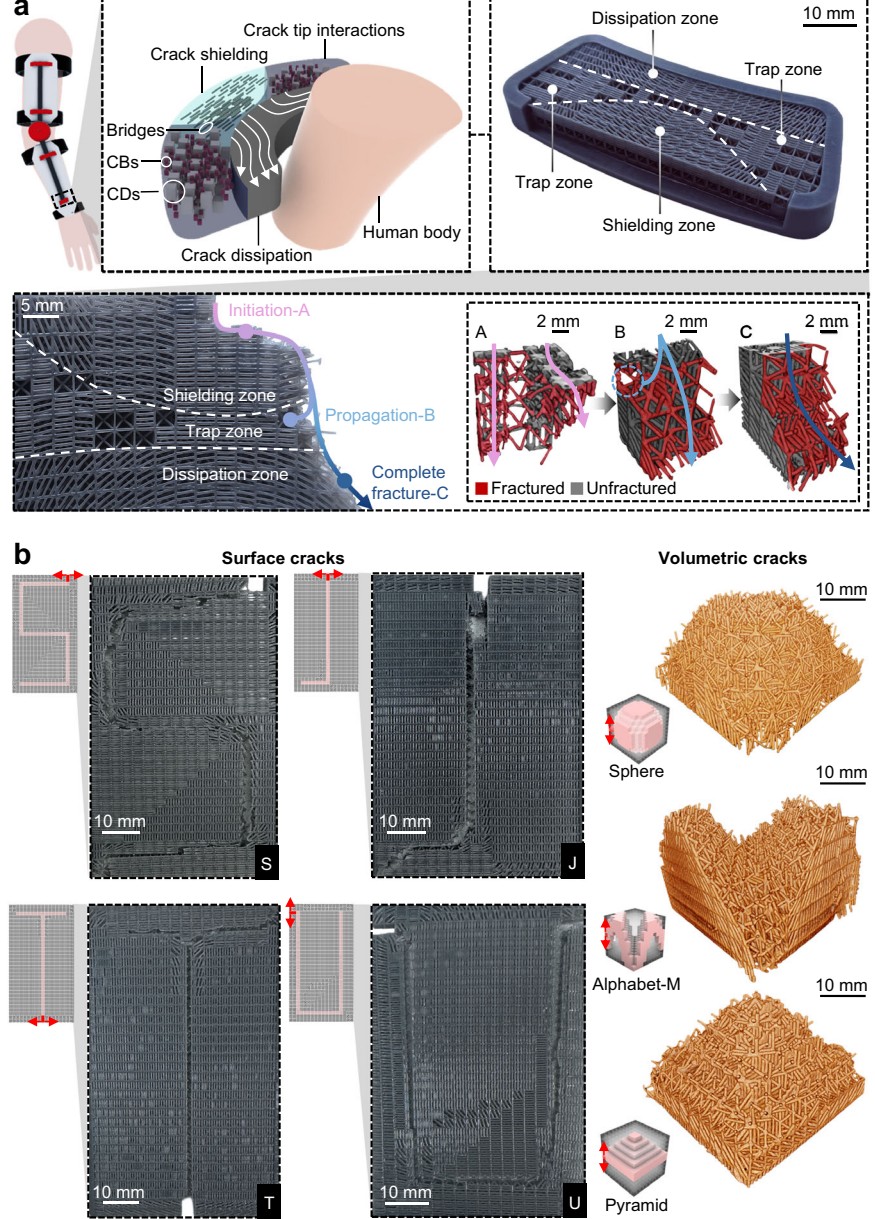

**Fig. 6 | DP metamaterials for functional applications. a** A personal protection suit incorporating the key toughening mechanisms, where CB and CD represents the crack bowing and crack deflection, respectively. The white arrow indicates the dissipation paths. **b** Functional cracks, including complex 2D alphabets (such as "SJTU") and 3D geometries such as a sphere, the alphabet-M, and a pyramid. Note that the 3D crack propagation diagrams in (**a**) and (**b**) are experimentally captured by X-ray computed tomography (XCT). The schematics images of the personal protection suit are modeled and rendered by Rhino 6. The images of the metamaterial samples are captured by camera.

beam elements. The reaction forces and displacements of the rigid rods and plates are recorded to obtain fracture load-displacement curves.

### Digital image correlation analysis

DIC measurements were conducted with experimental configuration provided in Supplementary Fig. 18 to measure the strain field during the deformation of the specimens with pre-engineered crack paths and crack-resisting mechanisms to interpret our results. The specimens were coated with white background color and scattered back dots as the reference points for the DIC calculations. Since a significant number of the features for the DP specimens are finer than the minimum unit of the DIC calculations, the spacing between adjacent structs were neglected for improved visualizations with continuous strain fields. The localized fracture strain in Fig. 2 was quantified by capturing strain data during the fracture process for the guiding cell and its corresponding

background cell with the same distance to the crack initiation point along the x axis, and calculating the fraction of fracture strain confined in the guiding cells. The fracture procedures of specimens were recorded with a camera with a 25 mm lens. The DIC images were captured with the subset resolution of 4000 × 3000 pixels, where the step dimensions are 1300 × 1400 pixels. The DIC analysis was performed with commercial software named Revealer-PMLGB 3D DIC, while a 7 × 7 pixels minimum size of the unit was applied during the DIC calculations.

### X-ray computed tomography analysis

XCT experiments were performed using the ZEISS Xradia 520 Versa X-ray microscope at the Instrumental Analysis Center of Shanghai Jiao Tong University. These experiments aimed to examine the internal fracture, in particular the interaction between pre-designed programming features with crack propagations. Fractured samples were

centrally placed within the X-ray zone. The experiments utilized a resolution range of 45.045 to 46.263 μm, and the field of view was set to 45 × 45 mm. The X-ray tomography was operated at 80 kV with a power setting of 7 W, using no filter. Each sample was rotated a full 360 degrees, capturing a total of 901 projection images during the scan, each with an exposure time of 1 sec. The projection images were reconstructed into slice images using the Control System Reconstructor software. The TXM 3D Viewer Software was employed to analyze the 3D crack propagation results, and the XMC Controller software was used to convert the projection images into slice images for detailed characterizations for the 3D fractures of tested samples.

## Data availability
The data supporting the findings of this study are included within the paper and its Supplementary Information. All other data are available from the corresponding authors upon request. Source data are provided with this paper.

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

## Acknowledgements

Yi Wu acknowledges the support from National Key Research and Development Program of China (No. 2023YFB3712001). Zhenyang Gao acknowledges the support from National Natural Science Foundation of China (523B2048). Hongze Wang acknowledge the support from National Natural Science Foundation of China (52075327 and 52004160), Shanghai Sailing Program (20YF1419200), Natural Science Foundation of Shanghai (20ZR1427500), SJTU Global Strategic Partnership Fund (2023 SJTU-CORNELL), the Innovation Foundation of Commercial Aircraft Manufacturing Engineering Center of China (No. 3-0410300-031), and the University Synergy Innovation Program of Anhui Province (GXXT-2022-086). Yang Lu acknowledges the support from the Research Grants Council of the Hong Kong Special Administrative Region, China, under RFS2021-1S05 and C7074-23G; Hong Kong RGC general research fund #11200623. The authors also acknowledge Ms. Tengteng Sun for obtaining the SEM images of metal composites.

## Author contributions

Z.G. conceptualized this study. Z.G., H.Z.W. and M.S.P. contributed to the methodology design of the study. Z.G. and X.Z. conducted data analysis, interpretation, and curation. Z.G. performed the experiments and wrote the manuscript. Z.G., H.Z.W., M.S.P., Y.L. and Y.W. substantially revised the work. Y.W., H.Z.W., C.X. and H.W.W. administered the project and provided supervision throughout the study.

## Competing interests

The authors declare no competing interests.
