## [Peer Review File · Nature Communications]

Damage-programmable design of metamaterials achieving crack-resisting mechanisms seen in natureREVIEWER COMMENTS

Reviewer #1 (Remarks to the Author):

This paper proposes a novel design of metamaterial for enhanced damage resilience and energy absorption. In general, though such thoughts have been flying among the researchers for a long time, this paper does a good job in implementing it through machine learning. However, some of the critical aspects that need serious attention are summarized below.

1. The prediction accuracy of the ML model needs more investigation. More results need to be added. Further the computational cost for forming it should be more clearly explained.
2. Most of the schematic figures are presented in the form of 3D lattices. However, crack propagation in such scenarios through a 3D space are not taken properly while presenting the results. The authors should keep things clear regarding such situations. In this context the machine learning models and corresponding validation results need to be furnished.
3. The rationale behind considering different building blocks in the materials should be explained more clearly (refer to supplementary figure 5a for example).
4. Aperiodic lattices have been proposed in the literature in conjunction with machine learning for enhanced strength and energy absorption (<https://doi.org/10.1016/j.jmps.2020.104112>, <https://doi.org/10.1016/j.actamat.2022.118226>, <https://doi.org/10.1016/j.addma.2020.101539>). Such works should be duly discussed. And the conceptual novelty needs to be highlighted more clearly.
5. Another major concern is the scenario of mixed mode loading (normal, shear or combined), or the cases when the load is applied in any arbitrary direction in the 3D space. This issue should be discussed with additional results to prove the generic nature of the current proposition.
6. The application figure is possibly redundant, and may be moved to supplementary material. Instead, more results figure should be added concerning ML model validation, applicability to 3D lattices and performance under complicated loading conditions.

Reviewer #3 (Remarks to the Author):

This study introduces a novel approach for developing damage-programmable metamaterials with pre-engineered microfibers in the cells, enabling spatial programming of microscale cracking behavior. Overall, the manuscript is well-

organized and well-written. The proposed strategy for producing metamaterials is intriguing. However, there are some suggestions and questions for the authors:

1. The primary contribution of the paper is stated as developing damage-programmable metamaterials. Upon careful examination, it raises the question: Is this the main contribution? It appears that a fixed microstructure of damage-programmable metamaterials may not be adaptable to various application scenarios. However, following the detailed design strategy presented in the paper, damage-programmable metamaterials can be customized. Therefore, it is worth considering if the primary contribution of this paper lies in proposing a systematic design strategy for damage-programmable metamaterials.

2. The direct correlation between natural materials and the proposed structures from Figure 1 is not clearly evident. Following the design strategy, it appears that the proposed damage-programmable metamaterials are mainly generated based on artificial design and mechanical analysis. The correlation between natural materials and the proposed metamaterials should be better clarified. Furthermore, the title should be revised since the proposed metamaterials are not found in nature.

3. The fracture energy-absorbing ability of the proposed metamaterials is determined by many parameters. Various theories and methods are adopted to determine the appropriate parameters for each cell. It may be possible to propose a unified global optimization strategy to determine these parameters, especially considering that machine learning algorithms have been adopted in the present paper.

4. In addition to comparing with the BCC structure, comparisons with other similar metamaterials should be provided to demonstrate the advancements of the proposed metamaterials.

5. Although application scenarios have been mentioned in the abstract and at the end of the paper, they remain at a conceptual level. Real application results should be provided to demonstrate the value of the proposed metamaterials, even if they are achieved in simplified scenarios or in the laboratory.

6. The manuscript contains an abundance of expressions in the active voice, such as 'we present', 'We use', 'we programmed', 'we demonstrated'. It is suggested that some of these be changed to the passive voice for more objective expressions. Additionally, the English expression should be improved.

Responses to Reviewers' comments

Reviewer #1 (Remarks to the Author):

This paper proposes a novel design of metamaterial for enhanced damage resilience and energy absorption. In general, though such thoughts have been flying among the researchers for a long time, this paper does a good job in implementing it through machine learning. However, some of the critical aspects that need serious attention are summarized below.

Response:

The authors are grateful for the Reviewer's positive feedback and valuable suggestions. The comments have significantly helped us in improving the manuscript. We have made careful revisions to each concern throughout the manuscript documents, highlighted in **yellow**, and referred by **blue**. Responses to each of the Reviewer's points follows below.

1. The prediction accuracy of the ML model needs more investigation. More results need to be added. Further the computational cost for forming it should be more clearly explained.

Response 1:

The author thanks the Reviewer for this valuable suggestion. During the training process, we allocated 10% of the training samples randomly to validate the accuracy of the machine learning (ML) model in this study, i.e. 10-fold cross-validation, adhering to a standard and common ML cross-validation practice. The initial manuscript reported the final prediction errors for fracture angle, fracture strength, and fracture energy to affirm the ML model's accuracy. The revised manuscript has included additional detailed data on ML prediction errors, showing the errors reduce and converge as the size of the training dataset increases, as shown in added **Supplementary Figure 4a, c, e**. To be more rigorous, the model's validity is further substantiated by experimental data from randomly generated DP cells, which show that the experimentally measured fracture properties of these cells fall within the error range of the model predictions, detailed in the added **Supplementary Figure 4b, d, f**. Associated texts are also carefully revised accordingly in the **Supplementary Information (lines 98-106 pages 4-5)** and referred from the **Main Manuscript (lines 107-110 page 5)**.

We addressed questions regarding computational cost of the proposed data-driven damage-programming metamaterial design method by detailing the time required for (1) gathering training data, (2) making ML predictions, and (3) generation of DP cells using ML predictions to search over vast design datasets. To be rigorous, all computational costs are evaluated using a 2nd Gen Intel® Core™ i5-12500H processor at 2.50 GHz and 16 GB of random access memory. The ML model is initially trained using the fracture data gathered from 900 training cells, taking 210 hours via finite element analysis (FEA). It's important to note that the computational expense of generating training data is a one-time cost and does

not recur during subsequent uses of the model. Therefore, the true computational cost for the data-driven damage-programming should primarily consider the cost associated with ML-assisted DP cell computation. The computational expense for ML fracture predictions of each DP cell is 34 μ s, averaged from recorded data up to 10^5 DP cells (added Supplementary Fig. 28a). This rapid computation enables the ML-assisted DP cell generation algorithm in this work, a task that is impractical with traditional numerical simulation approaches that takes 19 minutes in average to complete a single DP cell calculation. The time complexity for generating optimized DP cells follows the order of $O(n \log_{1/\beta} \pi / \theta_{min})$, where θ_{min} is the resolution of the face-fiber angles, β is the refinement coefficient, and n is the number of DP cells generated in each search cycle. The generation of a DP metamaterial structure by optimizing 10^5 DP cells with an angle resolution of 2° , can be achieved in 12 minutes (added Supplementary Fig. 28b). Additional contents are added in the Supplementary Information (lines 127-150 pages 6-7) and referred from the Main Manuscript (lines 107-110 page 5) to carefully discuss these computational costs.

2. Most of the schematic figures are presented in the form of 3D lattices. However, crack propagation in such scenarios through a 3D space are not taken properly while presenting the results. The authors should keep things clear regarding such situations. In this context the machine learning models and corresponding validation results need to be furnished.

Response 2:

We agree with the Reviewer that it is better to observe the 3D crack propagation to validate the data-driven damage-programming results as shown in schematics in a clearer manner (e.g., Fig. 3a). X-ray computed tomography (XCT) experiments and results are provided to characterize the resisted 3D crack propagations in revised Fig. 3a and discussed by revised texts in lines 140-144 page 6 in the Main Manuscript, clearly validating the schematics for the mechanisms of different toughening building blocks. The results in Fig. 4 are also added with XCT results (added Fig. 4c, revised texts in lines 226-229 page 10 and lines 241-245 page 11 in the Main Manuscript) to better compare the 3D crack propagations for metamaterials with and without damage-programming. Additional XCT-scanned 3D cracks have also been added to the application figure (revised Fig. 6) to further enrich the validations with 3D crack propagation visualizations. Detailed description of the XCT experiments is also provided in added texts in the Main Manuscript (lines 346-359 pages 15-16).

As the fast prediction tool used in data-driven damage-programming, the ML model is also validated with more data regarding the detailed model errors as the training data size increased, complemented by experimental validation results that further confirm the accuracy of the trained model (please refer to our Response 1).

3. The rationale behind considering different building blocks in the materials should be explained more clearly (refer to supplementary figure 5a for example).

Response 3:

We are sorry for potential confusions for the designs of different toughening building blocks. In clarify, the CB phase, CD phase, shield units, and bridges are designed using T0 to T3 DP cells to exhibit crack-resisting behaviors akin to phenomena such as crack bowing, crack deflection, crack shielding, and reinforcement bridging, which are commonly observed in materials from nature. A revision was made to clearly describe the detailed design motivation and rationales for each building blocks (lines 200-211 page 11 in Supplementary Note 3 and Supplementary Fig. 7 that was originally Supplementary Fig. 5). We also revised lines 85-91 page 4, lines 136-140 page 6, and lines 603-605 page 27 in the Main Manuscript to guide the author to these design rationales at the first mention of these building blocks.

4. Aperiodic lattices have been proposed in the literature in conjunction with machine learning for enhanced strength and energy absorption

(<https://doi.org/10.1016/j.jmps.2020.104112>, <https://doi.org/10.1016/j.actamat.2022.118226>, <https://doi.org/10.1016/j.addma.2020.101539>). Such works should be duly discussed. And the conceptual novelty needs to be highlighted more clearly.

Response 4:

We appreciate the Reviewer's suggestions. The literatures (including <https://doi.org/10.1016/j.jmps.2020.104112>, <https://doi.org/10.1016/j.actamat.2022.118226>) have reported interesting results in designing the lattice structures with programmable deformation sequences through anisotropic cell design based on machine learning or various design strategy to improve the compressive mechanical behaviors. The contributions of these works are properly cited in the revised manuscript (lines 46-50 page 2 in Main Manuscript). We revised our manuscript to highlight the novelty of our study that is the programmability of damage and fractures which are phenomena beyond the deformation. Because damage and fracture are complex, stochastic, and possess potentially catastrophic nature, it is very important to have a high controllability and programmability of damage. This makes the present study of programmable fractures in metamaterials an exciting area of research.

Some of the literature (<https://doi.org/10.1016/j.addma.2020.101539>) attempted to deflect cracks by introducing vacancies or weakening certain regions of lattices. However, precise control and understanding of fracture properties at the single-cell level remain largely unachieved, leading to no universally applicable strategies for damage-programming and fracture toughening in metamaterials proposed to date. Our work utilized the predictive power of machine learning to introduce the data-driven damage-programmable cell, marking a significant advancement in the design of metamaterials capable of controlling complex 3D fractures. This is also the first time that crack-resisting and programming mechanisms have been systematically adapted from nature to metamaterials, a development that has not been previously reported in the literature. (reflected in lines 55-63, page 3, and lines 75-78 pages 3-4 in Main Manuscript). The unique

contribution of this work provides a sound foundation for future designs of metamaterials with ability to control fractures and implements pre-programmed toughening mechanisms. This offers both experimental and theoretical frameworks to address the long-standing challenge of fractures in mechanical metamaterials.

5. Another major concern is the scenario of mixed mode loading (normal, shear or combined), or the cases when the load is applied in any arbitrary direction in the 3D space. This issue should be discussed with additional results to prove the generic nature of the current proposition.

Response 5:

We appreciate the Reviewer's suggestion to explore more complex load cases, which indeed presents an interesting engineering topic worth future investigation. Fractures in materials cover a broad research area and can occur due to various modes: mode I (normal opening), mode II (in-plane shear), mode III (out-of-plane shear), and mixed modes, with mode I being the most severe, fundamental, and commonly studied type of fracture. This paper is the first of its kind proposing and providing a sound foundation of a data-driven design to precisely program the damage in metamaterials. To clearly convey the main message and demonstrate the effectiveness of the damage-programming design approach, we focused the mode I. Nevertheless, we acknowledge that studying the effects of more complex load cases is important to provide more insights and valuable directions for future engineering research building on this work. In fact, the design of compact tension tests done in this study allows us to study this to a certain extent, in particular examining the behaviour of meta-structure in complex multi-axial stress states (which were present in front of existing cracks, in particular if we go down to the stress states each struts experiences), providing insights into how metamaterials designed in this study was able to control the fracture under complex stress states (see Fig. 4). We also made discussion on the future needs of carrying out more complex loading conditions in lines 273-282 pages 12-13 in the conclusions. Given the broad scope of fracture mechanics and its many facets for exploration, an exciting aspect of this study is to open up new opportunities for future research directions. Besides different fracture modes, this could include programming damage and developing crack-resisting mechanisms in various parent materials or topologies at different temperatures, and strain rates.

For the concern about the generality, the authors would like to kindly clarify that the data-driven design strategy is highly applicable and readily translatable to the fracture responses under other fracture modes. This extension is possible by incorporating additional training data that captures desired load directions, as the fracture properties in the fully connected neural networks (FCNNs) can represent responses from various fracture load scenarios. To demonstrate this, Supplementary Fig. 22 has been provided with additional discussions in lines 58-69 page 3 in Supplementary Information, showcasing how the proposed method could be expanded to other loading scenarios. This figure illustrates how different training data corresponding to additional load cases can be incorporated to expand the model's prediction and design. Since the training fracture properties fed into

the FCNNs can be derived from fracture data of any fracture modes, the validation in this work underlines the versatility of our method to accommodate different fracture types sets the stage for future engineering research, as elaborated in the revised paper.

6. The application figure is possibly redundant, and may be moved to supplementary material. Instead, more results figure should be added concerning ML model validation, applicability to 3D lattices and performance under complicated loading conditions.

Response 6:

We apologize for the potential redundant contents. The results in the application figure (Fig. 6) were rearranged, and the redundant part was moved to Supplementary Fig. 20. The texts in the Main Manuscript (lines 254-260 pages 11-12) are also revised accordingly. Detailed validation results of ML model were added in the revised manuscript (refer to Response 1).

More crack propagation results in 3D lattices and corresponding discussions are also provided in Fig. 3a, Fig. 4c, and Fig. 6b through additional XCT experiments (refer to Response 2). Discussions and additional contents have been made in the revised manuscript to discuss the potential expansion of the model to handle more complicated load cases that can be used in future research. Additionally, the revised manuscript identifies expansion potential of the proposed method for arbitrary load case, and clarified the rationale for using mode I fracture configuration throughout the study: to concisely demonstrate the main contribution of proposing a data-driven damage-programming design method that enables metamaterials with mechanisms for crack dissipation and resistance inspired by nature (refer to Response 5).

Reviewer #3 (Remarks to the Author):

This study introduces a novel approach for developing damage-programmable metamaterials with pre-engineered microfibers in the cells, enabling spatial programming of microscale cracking behavior. Overall, the manuscript is well-organized and well-written. The proposed strategy for producing metamaterials is intriguing. However, there are some suggestions and questions for the authors:

Response:

We thank the Reviewer for the positive feedback and the supportive suggestions. The suggestions have been instrumental in improving our manuscript. We have diligently reviewed and incorporated all comments, where the revised contents are highlighted in yellow and referred by blue. We have also prepared itemized responses to each point from the Reviewer below.

1. The primary contribution of the paper is stated as developing damage-programmable metamaterials. Upon careful examination, it raises the question: Is this the main contribution? It appears that a fixed microstructure of damage-

programmable metamaterials may not be adaptable to various application scenarios. However, following the detailed design strategy presented in the paper, damage-programmable metamaterials can be customized. Therefore, it is worth considering if the primary contribution of this paper lies in proposing a systematic design strategy for damage-programmable metamaterials.

Response 7:

The authors concur with the Reviewer that developing a systematic design strategy for damage-programmable metamaterials is the primary contributions of this paper. Accordingly, the lines 21-24 page 1 in abstract and lines 59-61 page 3 in introduction is carefully revised accordingly based on this suggestion. The title of this paper is also revised to “Damage-programmable design of metamaterials achieving crack-resisting mechanisms in nature”.

2.The direct correlation between natural materials and the proposed structures from Figure 1 is not clearly evident. Following the design strategy, it appears that the proposed damage-programmable metamaterials are mainly generated based on artificial design and mechanical analysis. The correlation between natural materials and the proposed metamaterials should be better clarified. Furthermore, the title should be revised since the proposed metamaterials are not found in nature.

Response 8:

We are sorry for not clearly showing the direct links between the proposed designs and toughening mechanisms found in natural materials. The connection between nature and damage-programmable (DP) metamaterials is shown via two key aspects. Firstly, similar to how materials in nature control the crack initiation and propagation thanks to internal micro-structure features, we have mimicked this strategy to develop metamaterials that can program the fracture behaviors thanks to internal unit cells at micro-scale level, as shown in the revised Fig. 1a.

Secondly, the link to nature is further established through the adoption of crack-resisting mechanisms such as crack tip interaction, crack shielding, and reinforcement bridging, illustrated in revised Fig. 1b (also refer to Response 3 for detailed supplementary design rationales). These mechanisms, commonly observed in ceramics, metals, and biological materials, were not realised in metamaterials before this research. One of the major contributions of this paper is successfully linking these natural crack-resisting mechanisms to the design of mechanical metamaterials. Revised texts in lines 79-91 page 4 in Main Manuscript and were made to better clarify this link. Since it is the design concept that links the DP metamaterials and nature, the title is revised accordingly to avoid confusions (refer to Response 7).

3.The fracture energy-absorbing ability of the proposed metamaterials is determined by many parameters. Various theories and methods are adopted to determine the appropriate parameters for each cell. It may be possible to propose a unified global optimization strategy to determine these parameters, especially

considering that machine learning algorithms have been adopted in the present paper.

Response 9:

The authors agree with the Reviewer to unify the design concepts linking different types of DP cells. Indeed, all the DP cells in this study have been optimized using a data-driven damage-programming metamaterial method, which generates 3D digital information of optimized design parameters for DP cells. In this work, these optimized parameters are informed by given damage designs determined by two key objectives (added Supplementary Fig. 1): spatial damage control, and the design of crack-resisting mechanisms inspired by nature.

To clearly convey this unified design concept, we revised Supplementary Note 1 and added Supplementary Fig. 1 as a unified design schematics figure to integrate the DP designs used throughout the paper. Other supplementary design details are also revised according to this modification. Furthermore, the corresponding text in the Main Manuscript (lines 82-85 page 4, lines 99-110 pages 4-5) has been carefully revised to ensure it accurately references this unified design approach.

4. In addition to comparing with the BCC structure, comparisons with other similar metamaterials should be provided to demonstrate the advancements of the proposed metamaterials.

Response 10:

We thank the Reviewer for the suggestions. In the previous manuscript, we compared the typical metamaterial topologies (BCC, FCC, octet-truss) exhibiting fracture characteristics similar to the base BCC topology of DP cells, using Ashby charts and property data in original Fig. 5. To enhance this comparison, we have now included additional topologies (vintiles, tesseract) that exhibit more ductile fracture behaviors in the updated Ashby charts and property data (revised Fig. 5). Additionally, Fig. 5b is also revised to include schematic images of the topologies to improve the readability of the chart.

5. Although application scenarios have been mentioned in the abstract and at the end of the paper, they remain at a conceptual level. Real application results should be provided to demonstrate the value of the proposed metamaterials, even if they are achieved in simplified scenarios or in the laboratory.

Response 11:

We appreciate the Reviewer's suggestions. Additional fracture testing results for representative regions of the application components with DP metamaterials have been added, where this focus on specific regions is necessitated by the build volume restrictions of the Form 3 3D printer. Specifically, a representative cross-section of the protective component depicted in Fig. 6a was fractured, and XCT results confirmed the functionality of each damage-programmed region. Since the application examples in original Fig. 6b-c both illustrate the spatial crack path

control function of the proposed design method, they have been moved to the newly added Supplementary Fig. 20 with an example of flying vehicles to prevent duplication (refer to Response 6), where representative sections of the flying vehicles with pre-engineered functional crack paths have been fabricated and tested to confirm the effectiveness of the proposed method. In main text, they have been replaced by the new Fig. 6b, which more broadly demonstrates the method's ability to program surface and volumetric crack responses using complex alphabets and shapes ("SJTU" for surface cracks, and sphere, 3D alphabet-M, and pyramid for volumetric cracks). The crack paths from these tests align with the design criteria. The corresponding discussions (lines 254-260 pages 11-12 in Main Manuscript) has been carefully updated to reflect these additions.

6. The manuscript contains an abundance of expressions in the active voice, such as 'we present', 'We use', 'we programmed', 'we demonstrated'. It is suggested that some of these be changed to the passive voice for more objective expressions. Additionally, the English expression should be improved.

Response 12:

We thank the Reviewer for the advice. The use of active voice has been reduced throughout the manuscript in response to the suggestion. A thorough proofreading has been conducted to enhance the language quality of the manuscript.

REVIEWERS' COMMENTS

Reviewer #1 (Remarks to the Author):

The paper has been carefully modified.

Reviewer #3 (Remarks to the Author):

The suggestions and questions raised last time have been properly addressed by the authors. Relevant content, such as comparisons with other metamaterials and the application of the proposed metamaterials, has also been supplemented and elaborated upon. Here are some minor suggestions for further improving the manuscript:

1. In line 43, it is suggested to revise "improved specific energy absorption" to "ultrahigh energy absorption" to remain consistent with the expression "ultrahigh stiffness-to-weight ratio." Additionally, various novel metamaterials have been systematically developed and proposed in recent research, demonstrating significantly improved energy absorption capabilities compared to conventional materials. Relevant research can be referred to:

- “Flexible, efficient and adaptive modular impact-resistant metamaterials”
- “Energy absorption of discretely assembled composite self-locked systems”

2. In most cases, the mechanical properties of cellular materials cannot be tuned on-demand after manufacture, and the crashworthiness of existing tailorable metamaterials is often limited. Thus, improving the ease of manufacture and flexibility is also critical to expanding the application scenarios for metamaterials, which may include applications in civil engineering. Therefore, the expression in line 44 may be more appropriately revised as:

“By combining desirable metamaterials with ease of manufacture and tailorability, these metamaterials could provide promising solutions to replace conventional materials in high-value applications such as biomedical industries, aerospace, and civil engineering.”

Relevant research on tailorable metamaterials can be referred to:

- “Mechanical and failure characteristics of novel tailorable architected metamaterials against crash impact”

Relevant research on the application of metamaterials in civil engineering can be referred to:

- “Study on Tamped Spherical Detonation-Induced Dynamic Responses of Rock and PMMA Through Mini-chemical Explosion Tests and a Four-Dimensional Lattice Spring Model”
- “Soil-water inrush induced shield tunnel lining damage and its stabilization: A case study”

3. Layered rocks also possess complex hierarchical features to resist fracture, which is suggested to be supplemented in Line 64. Relevant research that reveals the fracturing and mechanical behavior of layered rocks can be referred to:

- “A combined weighted Voronoi tessellation and random field approach for modeling heterogeneous rocks with correlated grain structure”

In general, this paper has been improved after the major revision and can be considered for acceptance after these minor revisions.

Responses to Reviewers' comments

Reviewer #1 (Remarks to the Author):

The paper has been carefully modified.

Response:

The authors would like to express our heartfelt gratitude to the Reviewer for the constructive feedback and positive assessment. Your insights have significantly enhanced the quality and scientific depth of this paper, and your encouragement has been invaluable in motivating us to continue our innovative research endeavors and contribute meaningfully to the scientific community.

Reviewer #3 (Remarks to the Author):

The suggestions and questions raised last time have been properly addressed by the authors. Relevant content, such as comparisons with other metamaterials and the application of the proposed metamaterials, has also been supplemented and elaborated upon.

Response:

The authors would like to sincerely appreciate the reviewer for their valuable suggestions and positive evaluations. Your insightful feedback has significantly helped us improve the manuscript and has been a great encouragement to us. We have made careful revisions in response to each comment throughout the manuscript documents, highlighted in yellow, and referred by blue. Our detailed responses to each of the reviewer's points are provided below.

Here are some minor suggestions for further improving the manuscript:

1. In line 43, it is suggested to revise "improved specific energy absorption" to "ultrahigh energy absorption" to remain consistent with the expression "ultrahigh stiffness-to-weight ratio." Additionally, various novel metamaterials have been systematically developed and proposed in recent research, demonstrating significantly improved energy absorption capabilities compared to conventional materials. Relevant research can be referred to:

"Flexible, efficient and adaptive modular impact-resistant metamaterials"
"Energy absorption of discretely assembled composite self-locked systems"

Response 1:

The authors thank the reviewer for the suggested terminology and useful references. We have replaced "improved specific energy absorption" with "ultrahigh energy absorption" and cited these novel studies ("*Flexible, efficient and adaptive modular impact-resistant metamaterials*" and "*Energy absorption of discretely assembled composite self-locked systems*") on line 45, page 2 of the Main Manuscript as suggested.

2. In most cases, the mechanical properties of cellular materials cannot be tuned on-demand after manufacture, and the crashworthiness of existing tailorable metamaterials is often limited. Thus, improving the ease of manufacture and flexibility is also critical to expanding the application scenarios for metamaterials, which may include applications in civil engineering. Therefore, the expression in line 44 may be more appropriately revised as:

“By combining desirable metamaterials with ease of manufacture and tailorability, these metamaterials could provide promising solutions to replace conventional materials in high-value applications such as biomedical industries, aerospace, and civil engineering.”

Relevant research on tailorable metamaterials can be referred to:

“Mechanical and failure characteristics of novel tailorable architected metamaterials against crash impact”

Relevant research on the application of metamaterials in civil engineering can be referred to:

“Study on Tamped Spherical Detonation-Induced Dynamic Responses of Rock and PMMA Through Mini-chemical Explosion Tests and a Four-Dimensional Lattice Spring Model”

“Soil-water inrush induced shield tunnel lining damage and its stabilization: A case study”

Response 2:

The authors thank the Reviewer for the suggested expression and innovative references, which has helped improve the literature review of this paper. We have replaced the original content of introduction with the suggested expression, and added those novel references to suggested positions of the sentence on lines 46-49, page 2 of the Main Manuscript (“Mechanical and failure characteristics of novel tailorable architected metamaterials against crash impact” at the end of ease of manufacture and tailorability, “Study on Tamped Spherical Detonation-Induced Dynamic Responses of Rock and PMMA Through Mini-chemical Explosion Tests and a Four-Dimensional Lattice Spring Model” and “Soil-water inrush induced shield tunnel lining damage and its stabilization: A case study” at the end of civil engineering).

3. Layered rocks also possess complex hierarchical features to resist fracture, which is suggested to be supplemented in Line 64. Relevant research that reveals the fracturing and mechanical behavior of layered rocks can be referred to:

“A combined weighted Voronoi tessellation and random field approach for modeling heterogeneous rocks with correlated grain structure”

Response 3:

The authors thank the Reviewer for pointing out the layered rocks, which are indeed an interesting category of innovative materials that uses hierarchical features against fracture. We have modified the original sentence to include the “layered rocks” with suggested reference (“A combined weighted Voronoi

tessellation and random field approach for modeling heterogeneous rocks with correlated grain structure”) on line 67, page 3 of the Main Manuscript.

In general, this paper has been improved after the major revision and can be considered for acceptance after these minor revisions.

Response:

The authors would like to once again express our sincere gratitude to the Reviewer for the insightful reviews and supportive feedback. The manuscript has been revised and improved as suggested in **Responses 1-3**.